# Improving Simple Models with Confidence Profiles

**Amit Dhurandhar\***
IBM Research,
Yorktown Heights, NY
adhuran@us.ibm.com

**Karthikeyan Shanmugam\***
IBM Research,
Yorktown Heights, NY
karthikeyan.shanmugam2@ibm.com

**Ronny Luss**
IBM Research,
Yorktown Heights, NY
rluss@us.ibm.com

**Peder Olsen**
IBM Research,
Yorktown Heights, NY
pederao@us.ibm.com
\*

## Abstract

In this paper, we propose a new method called ProfWeight for transferring information from a pre-trained deep neural network that has a high test accuracy to a simpler interpretable model or a very shallow network of low complexity and a priori low test accuracy. We are motivated by applications in interpretability and model deployment in severely memory constrained environments (like sensors). Our method uses linear probes to generate confidence scores through flattened intermediate representations. Our transfer method involves a theoretically justified weighting of samples during the training of the simple model using confidence scores of these intermediate layers. The value of our method is first demonstrated on CIFAR-10, where our weighting method significantly improves (3-4%) networks with only a fraction of the number of Resnet blocks of a complex Resnet model. We further demonstrate operationally significant results on a real manufacturing problem, where we dramatically increase the test accuracy of a CART model (the domain standard) by roughly 13%.

## 1 Introduction

Complex models such as deep neural networks have shown remarkable success in applications such as computer vision, speech and time series analysis [15, 18, 26, 10]. One of the primary concerns with these models has been their lack of transparency which has curtailed their widespread use in domains where human experts are responsible for critical decisions [21]. Recognizing this limitation, there has been a surge of methods recently [29, 27, 5, 28, 25] to make deep networks more interpretable. These methods highlight important features that contribute to the particular classification of an input by a deep network and have been shown to reasonably match human intuition.

We, in this paper, however, propose an intuitive model agnostic method to enhance the performance of simple models (viz. lasso, decision trees, etc.) using a pretrained deep network. A natural question to ask is, given the plethora of explanation techniques available for deep networks, why do we care about enhancing simple models? Here are a few reasons why simple models are still important.

**Domain Experts Preference:** In applications where the domain experts are responsible for critical decisions, they usually have a favorite model (viz. lasso in medical decision trees in advanced manufacturing) that they trust [31]. Their preference is to use something they have been using for years and are comfortable with.

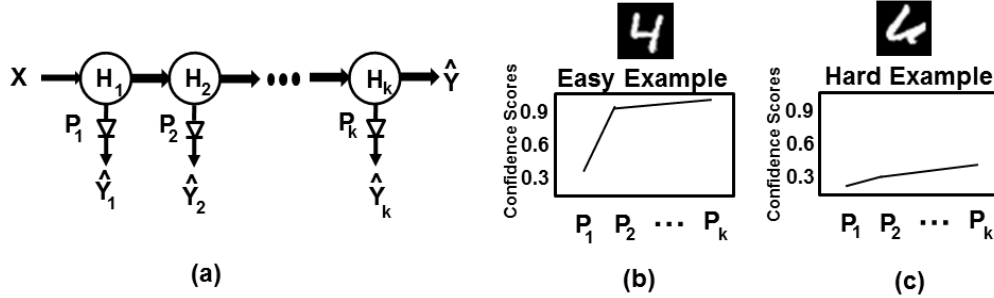

Figure 1: Above we depict our general idea. In (a), we see the $k$ hidden representations $H_1 \cdots H_k$ of a pretrained neural network. The diode symbols (triangle with vertical line) attached to each $H_i$ $\forall i \in \{1, ..., k\}$ denote the probes $P_i$ as in [3], with the $\hat{Y}_i$ denoting the respective outputs. In (b)-(c), we see example plots created by plotting the confidence score of the true label at each probe. In (b), we see a well written digit "4" which possibly is an easy example to classify and hence the confidence scores are high even at lower level probes. This sharply contrasts the curve in (c), which is for a much harder example of digit "4".

**Small Data Settings:** Companies usually have limited amounts of usable data collected for their business problems. As such, simple models are many times preferred here as they are less likely to overfit the data and in addition can provide useful insight [20]. In such settings, improving the simple models using a complex model trained on a much larger publicly/privately available corpora with the same feature representation as the small dataset would be highly desirable.

**Resource-Limited Settings:** In addition to interpretability, simple models are also useful in settings where there are power and memory constraints. For example, in certain Internet-of-Things (IoT) [24] such as those on mobile devices and in unmanned aerial vehicles (UAV) [11] there are strict power and memory constraints which preclude the deployment of large deep networks. In such settings, neural networks with only a few layers and possibly up to a few tens of thousands of neurons are considered reasonable to deploy.

We propose a method where we add probes to the intermediate layers of a deep neural network. A probe is essentially a logistic classifier (linear model with bias followed by a softmax) added to an intermediate layer of a pretrained neural network so as obtain its predictions from that layer. We call them *linear probes* throughout this paper. This is depicted in figure 1(a), where $k$ probes are added to $k$ hidden layers. Note that there is no backpropagation of gradients through the probes to the hidden layers. In other words, the hidden representations are fixed once the neural network is trained with only the probes being updated to fit the labels based on these previously learned representations. Also note that we are not required to add probes to each layer. We may do so only at certain layers which represent logical units for a given neural network architecture. For example, in a Resnet [18] we may add probes only after each Residual unit/block.

The confidence scores of the true label of an input when plotted at each of the probe outputs form a curve that we call a *confidence profile* for that input. This is seen in figure 1 (b)-(c). We now want to somehow use these confidence profiles to improve our simple model. It's worth noting that probes have been used before, but for a different purpose. For instance in [3], the authors use them to study properties of the neural network in terms of its stability and dynamics, but not for information transfer as we do. We consider functions of these confidence scores starting from an intermediate layer up to the final layer to weight samples during training of the simple model. The first function we consider, area under the curve (AUC) traced by the confidence scores, shows much improvement in the simple models performance. We then turn to learning the weights using neural networks that take as input the confidence scores and output an optimal weighting. Choice of the intermediate layers is chosen based on the simple model's complexity as is described later.

We observe in experiments that our proposed method can improve performance of simple models that are desirable in the respective domains. On CIFAR [19], we improve a simple neural network with very few Resnet blocks which can be deployed on UAVs and in IoT applications where there are memory and power constraints [24]. On a real manufacturing dataset, we significantly improve

---
**Algorithm 1** ProfWeight
---

**Input:** $k$ unit neural network $\mathcal{N}$, learning algorithm for simple model $\mathcal{L}_\mathcal{S}$, dataset $D_N$ used to train $\mathcal{N}$, dataset $D_S = \{(x_1, y_1), ..., (x_m, y_m)\}$ to train a simple model and margin parameter $\alpha$.

1) Attach probes $P_1, ..., P_k$ to the $k$ units of $\mathcal{N}$.
2) Train probes based on $D_N$ and obtain errors $e_1, ..., e_k$ on $D_S$. {There is no backpropagation of gradients here to the hidden units/layers of $\mathcal{N}$.}
3) Train the simple model $\mathcal{S} \leftarrow \mathcal{L}_\mathcal{S}(D_S, \beta, \vec{1}_m)$ and obtain its error $e_S$.{$\mathcal{S}$ is obtained by un-weighted training. $\vec{1}_m$ denotes a $m$ dimensional vector of 1s.}
4) Let $I \leftarrow \{u \mid e_u \leq e_S - \alpha\}$\{$I$ contains indices of all probes that are more accurate than the simple model $\mathcal{S}$ by a margin $\alpha$ on $D_S$.}
5) Compute weights $w$ using Algorithm 2 or 3 for AUC or neural network, respectively.
6) Obtain simple model $\mathcal{S}_{\mathbf{w},\beta} \leftarrow \mathcal{L}_{\mathcal{S},\beta}(D_S, \beta, w)$ {Train the simple model on $D_S$ along with the weights $w$ associated with each input.}
**return** $\mathcal{S}_{w,\beta}$

---

---
**Algorithm 2** AUC Weight Computation
---

**Input:** Neural network $\mathcal{N}$, probes $P_u$, dataset $D_S$, and index set $I$ from Algorithm 1.

1) Set $i \leftarrow 1$, $w = \vec{0}_m$ ($m$-vector of zeros)
**while** $i \leq m$ **do**
    2) Obtain confidence scores $\{c_{iu} = P_u(R_u(x_i))[y_i] \mid u \in I\}$.
    3) Compute $w_i \leftarrow \frac{1}{|I|} \sum_{u \in I} c_{iu}$ {In other words, estimate AUC for sample $(x_i, y_i) \in D_S$ based on probes indexed by $I$. $|\cdot|$ denotes cardinality.}
    4) Increment $i$, i.e., $i \leftarrow i + 1$
**end while**
**return** $w$

---

a decision tree classifier which is the method of choice of a fab engineer working in an advanced manufacturing plant.

The primary intuition behind our approach is to identify examples that the simple model will most likely fail on, i.e. identify *truly hard* examples. We then want to inform the simple model to ignore these examples and make it expend more effort on other *relatively easier* examples that it could perform well on, with the eventual hope of improving generalization performance. This is analogous to a teacher (i.e. complex model) informing a student (i.e. simple model) about aspects of the syllabus he should focus on and which he/she could very well ignore since it is way above his/her current level. We further ground this intuition and justify our approach by showing that it is a specific instance of a more general procedure that weights examples so as to learn the optimal simple model.

## 2 General Framework

In this section we provide a simple method to transfer information from a complex model to a simple one by characterizing the hardness of examples. We envision doing this with the help of confidence profiles that are obtained by adding probes to different layers of a neural network.

As seen in figure 1(b)-(c), our intuition is that easy examples should be resolved by the network, that is, classified correctly with high confidence at lower layer probes themselves, while hard examples would either not be resolved at all or be resolved only at or close to the final layer probes. This captures the notion of the network having to do more work and learn more finely grained representations for getting the harder examples correctly classified. One way to capture this notion is to compute the area under the curve (AUC) traced by the confidence scores at each of the probes for a given input-output pair. AUC amounts to averaging the values involved. Thus, as seen in figure 1(b), the higher the AUC, the easier is the example to classify. Note that the confidence scores are for the *true* label of that example and not for the predicted label, which may be different.

---

**Algorithm 3** Neural Network Weight Computation

---

**Input:** Weight space $\mathcal{C}$, dataset $D_S$, # of iterations $N$ and index set $I$ from Algorithm 1.

1) Obtain confidence scores $\{c_{iu} = P_u(R_u(x_i))[y_i] \mid u \in I\}$ for $x_i$ when predicting the class $y_i$ using the probes $P_u$ for $i \in \{1, \ldots, m\}$.
2) Initialize $i = 1$, $w^0 = \vec{1}_m$ and $\beta^0$ (simple model parameters)
**while** $i \leq N$ **do**
    3) Update simple model parameters: $\beta^i = \operatorname{argmin}_\beta \sum_{j=1}^m \lambda(S_{w^{i-1},\beta}(x_j), y_j)$
    4) Update weights: $w^i = \operatorname{argmin}_{w \in \mathcal{C}} \sum_{j=1}^m \lambda(S_{w,\beta^i}(x_j), y_j) + \gamma \mathcal{R}(w)$, where $\mathcal{R}(\cdot)$ is a regularization term set to $(\frac{1}{m}\sum_{i=1}^m w_i - 1)^2$ with scaling parameter $\gamma$. {Note that the weight space $\mathcal{C}$ restricts $w$ to be a neural network that takes as input the confidence scores $c_{iu}$}
    5) Increment $i$, i.e., $i \leftarrow i + 1$
**end while**
**return** $w = w^N$

---

We next formalize this intuition which suggests a truly hard example is one that is more of an outlier than a prototypical example of the class that it belongs to. In other words, if $X \times Y$ denotes the input-output space and $p(x, y)$ denotes the joint distribution of the data, then a hard example $(x_h, y_h)$ has low $p(y_h|x_h)$.

A learning algorithm $\mathcal{L}_S$ is trying to learn a simple model that "best" matches $p(y|x)$ so as to have low generalization error. The dataset $D_S = \{(x_1, y_1), ..., (x_m, y_m)\}$, which may or may not be representative of $p(x, y)$, but which is used to produce the simple model, may not produce this best match. We thus have to bias $D_S$ and/or the loss of $\mathcal{L}_S$ so that we produce this best match. The most natural way to bias is by associating *weights* $W = \{w_1, ..., w_m\}$ with each of the $m$ examples $(x_1, y_1), ..., (x_m, y_m)$ in $D_S$. This setting thus seems to have some resemblance to covariate shift [1] where one is trying to match distributions. Our goal here, however, is not to match distributions but to bias the dataset in such a way that we produce the best performing simple model.

If $\lambda(., .)$ denotes a loss function, $w$ a vector of $m$ weights to be estimated for examples in $D_S$, and $S_{w,\beta} = \mathcal{L}_S(D_S, \beta, w)$ is a simple model with parameters $\beta$ that is trained by $\mathcal{L}_S$ on the weighted dataset. $\mathcal{C}$ is the space of allowed weights based on constraints (viz. penalty on extremely low weights) that would eliminate trivial solutions such as all weights being close to zero, and $\mathcal{B}$ is the simple model's parameter space. Then ideally, we would want to solve the following optimization problem:

$$S^* = S_{w^*,\beta^*} = \min_{w \in \mathcal{C}} \min_{\beta \in \mathcal{B}} E\left[\lambda\left(S_{w,\beta}(x), y\right)\right] \tag{1}$$

That is, we want to learn the optimal simple model $S^*$ by estimating the corresponding optimal weights $W^*$ which are used to weight examples in $D_S$. It is known that not all deep networks are good density estimators [2]. Hence, our method does not just rely on the output confidence score for the true label, as we describe next.

## 2.1 Algorithm Description

We first train the complex model $\mathcal{N}$ on a data set $D_N$ and then freeze the resulting weights. Let $\mathcal{U}$ be the set of logical units whose representations we will use to train probes, and let $R_u(x)$ denote the flattened representation after the logical unit $u$ on input $x$ to the trained network $\mathcal{N}$. We train *probe function* $P_u(\cdot) = \sigma(Wx + b)$, where $W \in \mathbf{k} \times |\mathbf{R_u(x)}|$, $b \in \mathbb{R}^k$, $\sigma(\cdot)$ is the standard softmax function, and $k$ is the number of classes, on the flattened representations $R_u(x)$ to optimize the cross-entropy with the labels $y$ in the training data set $D_N$. For a label $y$ among the class labels, $P_u(R_u(x))[y] \in [0, 1]$ denotes the confidence score of the probe on label $y$.

Given that the simple model may have a certain performance, we do not want to use very low level probe confidence scores to convey hardness of examples to it. A teacher must be at least as good as the student and so we compute weights in Algorithm 1 only based on those probes that are roughly more accurate than the simple model. We also have parameter $\alpha$ which can be thought of as a margin parameter determining how much better the weakest teacher should be. The higher the $\alpha$, the better

the worst performing teacher will be. As we will see in the experiments, it is not always optimal to only use the best performing model as the teacher, since, if the teacher is highly accurate all confidences will be at or near 1 which will provide no useful information to the simple student model.

Our main algorithm, ProfWeight [2] is detailed in Algorithm 1. At a high level it can be described as performing the following steps:

- Attach and train probes on intermediate representations of a high performing neural network.

- Train a simple model on the original dataset.

- Learn weights for examples in the dataset as a function (AUC or neural network) of the simple model and the probes.

- Retrain the simple model on the final weighted dataset.

In step (5), one can compute weights either as the AUC (Algorithm 2) of the confidence scores of the selected probes or by learning a regularized neural network (Algorithm 3) that inputs the same confidence scores. In Algorithm 3, we set the regularization term $\mathcal{R}(w) = (\frac{1}{m} \sum_{i=1}^{m} w_i - 1)^2$ to keep the weights from all going to zero. Also as is standard practice when training neural networks [15], we also impose an $\ell_2$ penalty on the weights so as to prevent them from diverging. Note that, while the neural network is trained using batches of data, the regularization is still a function of all training samples. Algorithm 3 alternates between minimizing two blocks of variables ($w$ and $\beta$). When the subproblems have solutions and are differentiable, all limit points of $(w_k, \beta_k)$ can be shown to be stationary points [16]. The final step of ProfWeight is to train the simple model on $D_S$ with the corresponding learned weights.

## 2.2 Theoretical Justification

We next provide a justification for the regularized optimization in Step 4 of Algorithm 3. Intuitively, we have a pre-trained complex model that has high accuracy on a test data set $D_{\text{test}}$. Consider the binary classification setting. We assume that $D_{\text{test}}$ has samples drawn from a uniform mixture of two class distributions: $P(\mathbf{x}|y = 0)$ and $P(\mathbf{x}|y = 1)$. We have another simple model which is trained on a training set $D_{\text{train}}$ and has a priori low accuracy on the $D_{\text{test}}$. We would like to modify the training procedure of the simple model such that the test accuracy could be improved.

Suppose, training the simple model on training dataset $D_{\text{train}}$ results in classifier $M$. We view this training procedure of simple models through a different lens: It is equivalent to the optimal classification algorithm trained on the following class distribution mixtures: $P_M(\mathbf{x}|y = 1)$ and $P_M(\mathbf{x}|y = 0)$. We refer to this distribution as $\tilde{D}_{\text{train}}$. If we knew $P_M$, the ideal way to bias an entry $(\mathbf{x}, y) \in \tilde{D}_{\text{train}}$ in order to boost test accuracy would be to use the following importance sampling weights $w(\mathbf{x}, y) = \frac{P(\mathbf{x}|y)}{P_M(\mathbf{x}|y)}$ to account for the covariate shift between $\tilde{D}_{\text{train}}$ and $D_{\text{test}}$. Motivated by this, we look at the following parameterized set of weights, $w_{M'}(\mathbf{x}, y) = \frac{P(\mathbf{x}|y)}{P_{M'}(\mathbf{x}|y)}$ for every $M'$ in the simple model class. We now have following result (proof can be found in the supplement):

**Theorem 2.1.** *If $w_{M'}$ corresponds to weights on the training samples, then the constraint $\mathbb{E}_{P_M(\mathbf{x}|y)}[w_{M'}(\mathbf{x}, y)] = 1$ implies that $w_{M'}(\mathbf{x}, y) = \frac{P(\mathbf{x}|y)}{P_M(\mathbf{x}|y)}$.*

It is well-known that the error of the performance of the best classifier (Bayes optimal) on a distribution of class mixtures is the total variance distance between them. That is:

**Lemma 2.2.** *[8] The error of the Bayes optimal classifier trained on a uniform mixture of two class distributions is given by:* $\min_{\theta} \sum \mathbb{D}[L_{\theta}(x, y)] = \frac{1}{2} - \frac{1}{2} D_{\text{TV}}(P(\mathbf{x}|y = 1), P(\mathbf{x}|y = 0))$ *where $L(\cdot)$ is the $0, 1$ loss function and $\theta$ is parameterization over a class of classifiers that includes the Bayes optimal classifier. $D_{\text{TV}}$ is the total variation distance between two distributions. $P(\mathbf{x}|y)$ are the class distributions in dataset $D$.*

From Lemma 2.2 and Theorem 2.1, where $\theta$ corresponds to the parametrization of the simple model, it follows that:

$$\min_{M',\theta \text{ s.t. } \mathbb{E}_{\tilde{D}_{\text{train}}}[w'_M]=1} \mathbb{E}_{\tilde{D}}[w_{M'}(\mathbf{x}, \mathbf{y}) L_\theta(\mathbf{x}, y)] = \frac{1}{2} - \frac{1}{2} D_{\text{TV}}(P(\mathbf{x}|y=1), P(\mathbf{x}|y=0)) \quad (2)$$

The right hand side is indeed the performance of the Bayes Optimal classifier on the test dataset $D_{\text{test}}$. The left hand side justifies the regularized optimization in Step 4 of Algorithm 3, which is implemented as a least squares penalty. It also justifies the min-min optimization in Equation 1, which is with respect to the weights and the parameters of the simple model.

## 3 Experiments

In this section we experiment on datasets from two different domains. The first is a public benchmark vision dataset named CIFAR-10. The other is a chip manufacturing dataset obtained from a large corporation. In both cases, we see the power of our method, $\mathrm{ProfWeight}$, in improving the simple model.

We compare our method with training the simple model on the original unweighted dataset (Standard). We also compare with Distillation [14], which is a popular method for training relatively simpler neural networks. We lastly compare results with weighting instances just based on the output confidence scores of the complex neural network (i.e. output of the last probe $P_k$) for the true label (ConfWeight). This can be seen as a special case of our method where $\alpha$ is set to the difference in errors between the simple model and the complex network.

We consistently see that our method outperforms these competitors. This showcases the power of our approach in terms of performance and generality, where the simple model may not be minimizing cross-entropy loss, as is usually the case when using Distillation.

### 3.1 CIFAR-10

We now describe our methods on the CIFAR-10 dataset [3]. We report results for multiple $\alpha$'s of our ProfWeight scheme including ConfWeight which is a special case of our method. Further model training details than appear below are given in the supplementary materials.

**Complex Model:** We use the popular implementation of the Residual Network Model available from the TensorFlow authors [4] where simple residual units are used (no bottleneck residual units are used). The complex model has 15 Resnet units in sequence. The basic blocks each consist of two consecutive $3 \times 3$ convolutional layers with either 64, 128, or 256 filters and our model has five of each of these units. The first Resnet unit is preceded by an initial $3 \times 3$ convolutional layer with 16 filters. The last Resnet unit is succeeded by an average pooling layer followed by a fully connected layer producing 10 logits, one for each class. Details of the 15 Resnet units are given in the supplementary material.

**Simple Models:** We now describe our simple models that are smaller Resnets which use a subset of the 15 Resnet units in the complex model. All simple models have the same initial convolutional layer and finish with the same average pooling and fully connected layers as in the complex model above. We have four simple models with 3, 5, 7, and 9 ResNet units. The approximate relative sizes of these models to the complex neural network are $1/5, 1/3, 1/2, 2/3$, correpondingly. Further details are about the ResNet units in each model are given in the supplementary material.

**Probes Used:** The set of units $\mathcal{U}$ (as defined in Section 2.1) whose representations are used to train the probes are the units in Table 1 of the trained complex model. There are a total of 18 units.

**Training-Test Split:** We split the available 50000 training samples from the CIFAR-10 dataset into training set 1 consisting of 30000 examples and training set 2 consisting of 20000 examples. We split the 10000 test set into a validation set of 500 examples and a holdout test set of 9500 examples. All final test accuracies of the simple models are reported with respect to this holdout test set. The validation set is used to tune all models and hyperparameters.

|            | SM-3            | SM-5           | SM-7            | SM-9            |
|------------|-----------------|----------------|-----------------|-----------------|
| Standard   | 73.15(± 0.7)    | 75.78(±0.5)    | 78.76(±0.35)    | 79.9(±0.34)     |
| ConfWeight | 76.27 (±0.48)   | 78.54 (±0.36)  | **81.46**(±0.50)| 82.09 (±0.08)   |
| Distillation | 65.84(±0.60)  | 70.09 (±0.19)  | 73.4(±0.64)     | 77.30 (±0.16)   |
| ProfWeight$^{\text{ReLU}}$ | **77.52** (±0.01) | 78.24(±0.01) | 80.16(±0.01) | 81.65 (±0.01) |
| ProfWeight$^{\text{AUC}}$ | 76.56 (±0.62) | **79.25**(±0.36) | **81.34**(±0.49) | **82.42** (±0.36) |

Table 1: Averaged accuracies (%) of simple model trained with various weighting methods and distillation. The complex model achieved $84.5\%$ accuracy. Weighting methods that average confidence scores of higher level probes perform the best or on par with the best in all cases. In each case, the improvement over the unweighted model is about $3 - 4\%$ in test accuracy. Distillation performs uniformly worse in all cases.

**Complex Model Training:** The complex model is trained on training set 1. We obtained a test accuracy of $0.845$ and keep this as our complex model. We note that although this is suboptimal with respect to Resnet performances of today, we have only used 30000 samples to train.

**Probe Training:** Linear probes $P_u(\cdot)$ are trained on representations produced by the complex model on training set 1, each for 200 epochs. The trained Probe confidence scores $P_u(R_u(x))$ are evaluated on samples in training set 2.

**Simple Models Training:** Each of the simpler models are trained only on training set 2 consisting of 20000 samples for 500 epochs. All training hyperparameters are set to be the same as in the previous cases. We train each simple model in Table 2 for the following different cases. Standard trains a simple unweighted model. ConfWeight trains a weighted model where the weights are the true label's confidence score of the complex model's last layer. Distillation trains the simple model using cross-entropy loss with soft targets obtained from the softmax ouputs of the complex model's last layer rescaled by temperature $t = 0.5$ (tuned with cross-validation) as in distillation of [14]. ProfWeight$^{\text{AUC}}$ and ProfWeight$^{\text{ReLU}}$ train using Algorithm 1 with Algorithms 2 and 3 for the weighting scheme, respectively. Results are for layer 14 as the lowest layer (margin parameter $\alpha$ was set small, and $\alpha = 0$ corresponded to layer 13). More details along with results for different temperature in distillation and margin in ProfWeight are given in the supplementary materials.

Test accuracies (their means and standard deviations each averaged over about 4 runs each) of each of the 4 simple models in Table 2 trained in 6 different ways described above are provided in Figure 3. Their numerical values in tabular form are given in Table 1.

**Results:** From Table 1, it is clear that in all cases, the weights corresponding to the AUC of the probe confidence scores from unit 13 or 14 and upwards are among the best in terms of test accuracies. They significantly outperform distillation-based techniques and, further, are better than the unweighted test accuracies by $3 - 4\%$. This shows that our $\mathrm{ProfWeight}$ algorithm performs really well. We notice that in this case, the confidence scores from the last layer or final probe alone are quite competitive as well. This is probably due to the complex model accuracy not being very high, having been trained on only 30000 examples. This might seem counterintuitive, but a highly accurate model will find almost all examples easy to classify at the last layer leading to confidence scores that are uniformly close to 1. Weighting with such scores then, is almost equivalent to no weighting at all. This is somewhat witnessed in the manufacturing example where the complex neural network had an accuracy in the 90s and ConfWeight did not enhance the CART model to the extent ProfWeight did. In any case, weighting based on the last layer is just a special instance of our method ProfWeight, which is seen to perform quite well.

## 3.2 Manufacturing

We now describe how our method not only improved the performance of a CART model, but produced operationally significant results in a semi-conductor manufacturing setting.

**Setup:** We consider an etching process in a semi-conductor manufacturing plant. The goal is to predict the quantity of metal etched on each wafer – which is a collection of chips – without having to explicitly measure it using high precision tools, which are not only expensive but also substantially slow down the throughput of a fab. If $T$ denotes the required specification and $\gamma$ the allowed variation, the target we want to predict is quantized into three bins namely: $(-\infty, T - \gamma), (T + \gamma, \infty)$ and

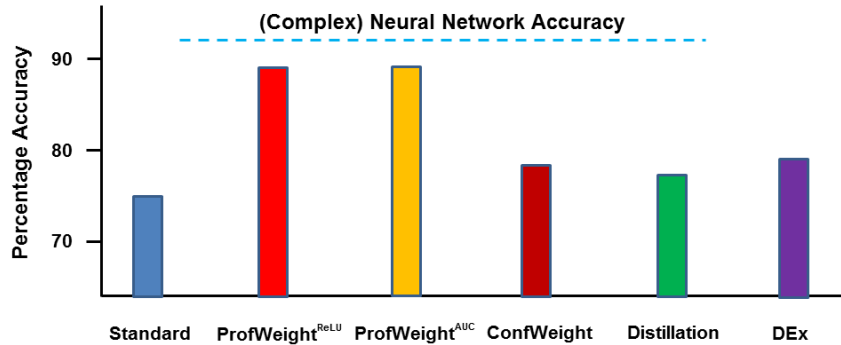

Figure 2: Above we show the performance of the different methods on the manufacturing dataset.

within spec which is $T \pm \gamma$. We thus have a three class problem and the engineers goal is not only to predict these classes accurately but also to obtain insight into ways that he can improve his process.

For each wafer we have 5104 input measurements for this process. The inputs consist of acid concentrations, electrical readings, metal deposition amounts, time of etching, time since last cleaning, glass fogging and various gas flows and pressures. The number of wafers in our dataset was 100,023. Since these wafers were time ordered we split the dataset sequentially where the first 70% was used for training and the most recent 30% was used for testing. Sequential splitting is a very standard procedure used for testing models in this domain, as predicting on the most recent set of wafers is more indicative of the model performance in practice than through testing using random splits of train and test with procedures such as 10-fold cross validation.

**Modeling and Results:** We built a neural network (NN) with an input layer and five fully connected hidden layers of size 1024 each and a final softmax layer outputting the probabilities for the three classes. The NN had an accuracy of 91.2%. The NN was, however, not the model of choice for the fab engineer who was more familiar and comfortable using decision trees.

Given this, we trained a CART based decision tree on the dataset. As seen in figure 2, its accuracy was 74.3%. Given the big gap in performance between these two methods the engineers wanted an improved interpretable model whose insights they could trust. We thus tested by weighting instances based on the actual confidence scores outputted by the NN and then retraining CART. This improved the CART performance slightly to 77.1%. We then used ProfWeight$^{AUC}$, where $\alpha$ was set to zero, to train CART whose accuracy bumped up significantly to 87.3%, which is a 13% lift. Similar gains were seen for ProfWeight$^{ReLU}$ where accuracy reached 87.4%. For Distillation we tried 10 different temperature scalings in multiples of 2 starting with 0.5. The best distilled CART produced a slight improvement in the base model increasing its accuracy to 75.6%. We also compared with the decision tree extraction (DEx) method [6] which had a performance of 77.5%.

**Operationally Significant Human Actions:** We reported the top features based on the improved model to the engineer. These features were certain pressures, time since last cleaning and certain acid concentrations. The engineer based on this insight started controlling the acid concentrations more tightly. This improved the total number of within spec wafers by 1.3%. Although this is a small number, it has huge monetary impact in this industry, where even 1% increase in yield can amount to billions of dollars in savings.

## 4 Related Work and Discussion

Our information transfer procedures based on confidence measures are related to Knowledge Distillation and learning with privileged information [22]. The key difference is in the way we use information. We weight training instances by functions, such as the average, of the confidence profiles of the training label alone. This approach, unlike Distillation [14, 26, 30], is applicable in broader settings like when target models are classifiers optimized using empirical risk (e.g., SVM) where risk could be any loss function. By weighting instances, our method uses any available target training methods. Distillation works best with cross entropy loss and other losses specifically designed for

this purpose. Typically, distilled networks are usually quite deep. They would not be interpretable or be able to respect tight resource constraints on sensors. In [24], the authors showed that primarily shallow networks can be deployed on memory constrained devices. The only papers to our knowledge that do thin down CNNs came about prior to ResNet and the memory requirements are higher even compared to our complex Resnet model (2.5 M vs 0.27 M parameters) [26].

It also interesting to note that calibrated scores of a highly accurate model does not imply good transfer. This is because post-calibration majority of the confidence scores would still be high (say >90%). These scores may not reflect the true hardness. Temperature scaling is one of the most popular methods for calibration of neural networks [17]. Distillation which involves temperature scaling showed subpar performance in our experiments.

There have been other strategies [9, 4, 6] to transfer information from bigger models to smaller ones, however, they are all similar in spirit to Distillation, where the complex models predictions are used to train a simpler model. As such, weighting instances also has an intuitive justification where viewing the complex model as a teacher and the TM as a student, the teacher is telling the student which aspects he/she should focus on (i.e. easy instances) and which he/she could ignore.

There are other strategies that weight examples although their general setup and motivation is different, for instance curriculum learning (CL) [7] and boosting [13]. CL is a training strategy where first easy examples are given to a learner followed by more complex ones. The determination of what is simple as opposed to complex is typically done by a human. There is usually no automatic gradation of examples that occurs based on a machine. Also sometimes the complexity of the learner is increased during the training process so that it can accurately model more complex phenomena. In our case however, the complexity of the simple model is assumed fixed given applications in interpretability [12, 23, 25] and deployment in resource limited settings [24, 11]. Moreover, we are searching for just one set of weights which when applied to the original input (not some intermediate learned representations) the fixed simple model trained on it gives the best possible performance. Boosting is even more remotely related to our setup. In boosting there is no high performing teacher and one generally grows an ensemble of weak learners which as just mentioned is not reasonable in our setting. Hard examples w.r.t. a previous 'weak' learner are highlighted for subsequent training to create diversity. In our case, hard examples are w.r.t. an accurate complex model. This means that these labels are near random. Hence, it is important to highlight these relatively easier examples when training the simple model.

In this work we proposed a strategy to improve simple models, whose complexity is fixed, with the help of a high performing neural network. The crux of the idea was to weight examples based on a function of the confidence scores based on intermediate representations of the neural network at various layers for the true label. We accomplished this by attaching probes to intermediate layers in order to obtain confidence scores. As observed in the experiments, our idea of weighting examples seems to have a lot of promise where we want to improve (interpretable) models trained using empirical risk minimization or in cases where we want to improve a (really) small neural network that will respect certain power and memory constraints. In such situations Distillation seems to have limited impact in creating accurate models.

Our method could also be used in small data settings which would be analogous to our setup on CIFAR 10, where the training set for the complex and simple models were distinct. In such a setting, we would obtain soft predictions from the probes of the complex model for the small data samples and use ProfWeight with these scores to weight the smaller training set. A complementary metric that would also be interesting to look at is the time (or number of epochs) it takes to train the simple model on weighted examples to reach the unweighted accuracy. If there is huge savings in time, this would be still useful in power constrained settings.

In the future, we would like to explore more adaptive schemes and hopefully understand them theoretically as we have done in this work. Another potentially interesting future direction is to use a combination of the improved simple model and complex model to make decisions. For instance, if we know that the simple models performance (almost) matches the performance of the complex model on a part of the domain then we could use it for making predictions for the corresponding examples and the complex model otherwise. This could have applications in interpretability as well as in speeding up inference for real time applications where the complex models could potentially be large.

## Acknowledgement

We would like to thank the anonymous area chair and reviewers for their constructive comments.

## Footnotes

\*First two authors have equal contribution

[2]Code is available at `https://github.ibm.com/Karthikeyan-Shanmugam2/Transfer/blob/master/README.md`

[3] We used the python version from https://www.cs.toronto.edu/ kriz/cifar.html.

[4] Code was obtained from: https://github.com/tensorflow/models/tree/master/research/resnet

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
