[Supplementary Material]

# Supplemental Material

This supplementary material contains additional tables, figures, and a proof.

## A    Additional Tables and Figures

| Units | Description | |
|---|---|---|
| Init-conv | $\begin{bmatrix} 3 \times 3 \text{ conv}, & 16 \end{bmatrix}$ | |
| Resunit:1-0 | $\begin{bmatrix} 3 \times 3 \text{ conv}, & 64 \\ 3 \times 3 \text{ conv}, & 64 \end{bmatrix}$ | |
| (Resunit:1-x)$\times$ 4 | $\begin{bmatrix} 3 \times 3 \text{ conv}, & 64 \\ 3 \times 3 \text{ conv}, & 64 \end{bmatrix}$ | $\times 4$ |
| (Resunit:2-0) | $\begin{bmatrix} 3 \times 3 \text{ conv}, & 128 \\ 3 \times 3 \text{ conv}, & 128 \end{bmatrix}$ | |
| (Resunit:2-x)$\times$ 4 | $\begin{bmatrix} 3 \times 3 \text{ conv}, & 128 \\ 3 \times 3 \text{ conv}, & 128 \end{bmatrix}$ | $\times 4$ |
| (Resunit:3-0) | $\begin{bmatrix} 3 \times 3 \text{ conv}, & 256 \\ 3 \times 3 \text{ conv}, & 256 \end{bmatrix}$ | |
| (Resunit:3-x)$\times$ 4 | $\begin{bmatrix} 3 \times 3 \text{ conv}, & 256 \\ 3 \times 3 \text{ conv}, & 256 \end{bmatrix}$ | $\times 4$ |
| Average Pool | | |
| Fully Connected - 10 logits | | |

Table 2: 18 unit Complex Model with 15 ResNet units used on CIFAR-10 experiments in Section 3.1

| Simple Model IDs | Additional Resunits | Rel. Size |
|---|---|---|
| SM-3 | None | $\approx 1/5$ |
| SM-5 | (Resunit:1-x)$\times$1 <br> (Resunit:2-x)$\times$1 | $\approx 1/3$ |
| SM-7 | (Resunit:1-x)$\times$2 <br> (Resunit:2-x)$\times$1 <br> (Resunit:3-x)$\times$1 | $\approx 1/2$ |
| SM-9 | (Resunit:1-x)$\times$2 <br> (Resunit:2-x)$\times$2 <br> (Resunit:3-x)$\times$2 | $\approx 2/3$ |

Table 3: Additional Resnet units in the Simple Models apart from the commonly shared ones. The last column shows the approximate size of the simple models relative to the complex neural network model in the previous table.

| Probes | 1 | 2 | 3 | 4 | 5 | 6 | 7 | 8 | 9 |
|---|---|---|---|---|---|---|---|---|---|
| Training Set 2 Accuracy | 0.298 | 0.439 | 0.4955 | 0.53855 | 0.5515 | 0.5632 | 0.597 | 0.6173 | 0.6418 |
| Probes | 10 | 11 | 12 | 13 | 14 | 15 | 16 | 17 | 18 |
| Training Set 2 Accuracy | 0.66104 | 0.6788 | 0.70855 | 0.7614 | 0.7963 | 0.82015 | 0.8259 | 0.84214 | 0.845 |

Table 4: Probes at various units and their accuracies on the training set 2 for the CIFAR-10 experiment. This is used in the $\mathrm{ProfWeight}^{\mathrm{AUC}}$ algorithm to choose the unit above which confidence scores needs to be averaged.

# B   Additional Training Details

**CIFAR-10 Experiments in Section 3.1**

**Complex Model Training:** We trained with an $\ell$-2 weight decay rate of $0.0002$, sgd optimizer with Nesterov momentum (whose parameter is set to 0.9), $600$ epochs and batch size $128$. Learning rates are according to the following schedule: $0.1$ till $40k$ training steps, $0.01$ between $40k$-$60k$ training steps, $0.001$ between $60k - 80k$ training steps and $0.0001$ for $> 80k$ training steps. This is the standard schedule followed in the code by the Tensorflow authors (code is taken from: https://github.com/tensorflow/models/tree/master/research/resnet). We keep the learning rate schedule invariant across all our results.

**Simple Models Training:**

1. **Standard**: We train a simple model as is on the training set 2.

2. **ConfWeight**: We weight each sample in training set 2 by the confidence score of the last layer of the complex model on the true label. As mentioned before, this is a special case of our method, ProfWeight.

3. **Distillation**: We train the simple model using a cross entropy loss with soft targets. Soft targets are obtained from the softmax ouputs of the last layer of the complex model (or equivalently the last linear probe) rescaled by temperature $t$ as in distillation of [14]. By using cross validation, we picked the temperature that performed best on the validation set in terms of validation accuracy for the simple models. We cross-validated over temperatures from the set $\{0.5, 3, 10.5, 20.5, 30.5, 40.5, 50\}$. See Figures 3 and 4 for validation and test accuracies for SM-9 model with distillation at different temperatures.

4. **ProfWeight**: Implementation of our ProfWeight algorithm where the weight of every sample in training set 2 is set to a function (depending on the choice of ReLu or AUC) of the probe confidence scores of the true label corresponding to units above the $14$-th unit. The rationale is that the probe precision at layer $14$ onwards are above the unweighted test scores of all the simple models in Table 4. The unweighted (i.e. Standard model) test accuracies from Table 1 can be checked against the accuracies of different probes on training set 2 given in Table 4 in the supplementary material.

Figure 3: Plot of validation set accuracy as a function of training steps for SM-9 simple model. The training is done using distillation. Validation accuracies for different temperatures used in distillation are plotted.

| Distillation Temperatures | Test Accuracy of SM-9 |
|---|---|
| 0.5 | 0.7719999990965191 |
| 3.0 | 0.709789470622414 |
| 5.0 | 0.7148421093037254 |
| 10.5 | 0.6798947390757109 |
| 20.5 | 0.7237894786031622 |
| 30.5 | 0.7505263184246264 |
| 40.5 | 0.7513684191201863 |
| 50 | 0.7268421022515548 |

Figure 4: Test Set accuracies of various versions of simple model SM-9 trained using distilled final layer confidence scores at various temperatures. The top two are for temperatures $0.5$ and $40.5$.

## C  Proof of Theorem 2.1

It is enough to show that, for two fixed distributions $P(x|y)$ and $P_M(x|y)$ with density functions $f(x|y)$ and $f_M(x|y)$: $\int \frac{f(x|y)f_M(x|y)}{r(x)} d(x) = 1$, $\int r(x) = 1$, $r(x) > 0$ $\forall x$ means that $r(x) = f(x|y)$ or $f_M(x|y)$. We show this for discrete distributions below.

**Lemma C.1.** *If $p$, $q$ and $r$ are three $n$ dimensional distributions then, $\sum_x \frac{p(x)r(x)}{q(x)} = 1$ only if either $q = p$ or $q = r$ pointwise.*

*Proof.* We first describe proofs for specific cases so as to provide some intuition about the general result.

If $p$, $r$ and $q$ are two dimensional distributions then if $\sum_{i=1,2} \frac{p_i r_i}{q_i} = 1$ we have,

$$\sum_{i=1,2} \frac{p_i r_i}{q_i} = 1$$

$$(q_1 + q_2) \sum_{i=1,2} p_i r_i = \prod_{i=1,2} q_i (p_1 + p_2)(r_1 + r_2)$$

$$(r_1 q_2 - r_2 q_1)(p_1 q_2 - p_2 r_1) = 0$$

This implies either $\frac{q_1}{q_2} = \frac{p_1}{p_2}$ or $\frac{q_1}{q_2} = \frac{r_1}{r_2}$. Without loss of generality (w.l.o.g.) assume $\frac{q_1}{q_2} = \frac{p_1}{p_2}$. Then $\frac{q_1}{q_2} + \frac{q_2}{q_2} = \frac{p_1}{p_2} + \frac{p_2}{p_2} \Rightarrow \frac{1}{q_2} = \frac{1}{p_2}$ or $p_2 = q_2$ which proves our result.

If $p = r$, then for $n$ dimensional distributions we have,

$$\sum_{i=1}^{n} \frac{p_i^2}{q_i} = 1$$

$$\sum_{i=1}^{n} \left( p_i^2 \prod_{j \neq i} q_j \right) = \prod_{i=1}^{n} q_i$$

$$\left( \sum_{i=1}^{n} q_i \right) \sum_{i=1}^{n} \left( p_i^2 \prod_{j \neq i} q_j \right) = \prod_{i=1}^{n} q_i \left( \sum_{i=1}^{n} p_i \right)^2$$

$$\sum_{i=1}^{n} \sum_{j=1, j \neq i}^{n} \prod_{k \neq i,j} q_k (p_i q_j - p_j q_i)^2 = 0$$

This implies that the polynomial is pointwise zero only if $\frac{p_i}{p_j} = \frac{q_i}{q_j}$ $\forall i, j$. This again gives our result of $p = q$.

For the general case analogous to previous results we get polynomials $(p_i q_j - p_j q_i)^2 (r_i q_j - r_j q_i)^2$ multiplied by positive constants that must be pointwise 0. Thus, $\frac{p_i}{p_j} = \frac{q_i}{q_j}$ or $\frac{r_i}{r_j} = \frac{q_i}{q_j}$. W.l.o.g. we can

assume that for half or more of the cases the ratio of $p_i$, $p_j$s are equal to the ratio of $q_i$, $q_j$s. In this case, only these equations can be considered along with constraints ensuring $p$ and $q$ are distributions and must sum to 1. Since the number of equations with ratios grow quadratically in the number of variables the hardest cases to show are when we have 4 (or fewer) variables. Using tools such as mathematica one can show that the other ratios also have to be equal or that $p = q$.

$\square$