[Reviews · NeurIPS 2018]

Reviewer 1



The authors present a solution for a difficult problem: how to use the information conveyed by a complex classifier, when practical circumstances require that a simple classifier is used? Now one might question why this is a problem at all, but the current development of very complex classifiers seems to suggest that more interpretable classifiers would be generally welcome as long as the accuracy penalty could be minimized. This is what the authors try here, by importing what seems to be an intuition behind curriculum learning: they feed an interpretable classifier with easy examples, where "easy" is indicated by the complex classifier. The authors show that this intuition works by presenting some empirical analysis that is very interesting and some theoretical analysis that I don't find very convincing. Mostly the proof of this idea is in empirical success in useful applications, for I could easily imagine an "intuition" that would point in exactly the opposite direction: that is, one might argue that the interpretable classifier should be fed with HARD examples that would more precisely specify the classification boundary. All in all I think the authors have succeeded in showing that, at least in some situations, their idea makes sense; more work is needed to actually understand when this idea works. The text is well written and flows quite well, even if it uses some (perhaps inevitable) jargon that ranges from boosting to probes and to many other notions. I think the presentation is quite effective overall. One point: the text mentions "Line 7" of Algorithm 3, but it would be better to simply say "Step 4". Also, I believe Ref. [4] is by "Caruana". And there are problems with capitalization in the references.

Reviewer 2



The authors introduce ProfWeight - a method for transferring knowledge from a teacher model to a student model. A "confidence profile" (taken from classification layers placed throughout the network) is used to determine which training samples are easy and which are hard. The loss function for the student model is weighted to favor learning the easier samples. The authors test this method on CIFAR10 and a real-world dataset. Quality: The idea presented by this paper is interesting and well-motivated. The method and results could be presented with more clarity, and the paper could benefit from some additional empirical analysis. However, overall the quality of the experiments are good. Clarity: Most of the section are straightforward. However, the method section is confusing and is not strategically organized. Additionally, there are some notational issues (see small notes). Specifically, the paper is missing a description of ProfWeight in the text. The components of the algorithm (weights, AUC/neural net) are described in Section 3, but there is no overview of the exact steps of the algorithm (learn a simple model, use error of simple model and the AUC/neural net to learn weights, retrain simple model with weights). A reader can only learn this overview from Algorithm 1, which is hard to parse because of all the notation. Similarly, Algorithm 3 should contain more description in the text. Originality: The idea of student/teacher transfer with sample weighting is novel and interesting. Significance: Model compression and interpretability is an important problem, and this paper presents a promising new method. It will be of interest to the community at large. Overall: I vote to accept this paper, since the method proposed is novel and seems promising. I do believe the methods section of this paper should be clarified, and more empirical results on other datasets would strengthen the paper. Detailed comments: Section 3.2 is a bit terse. The left-hand side of equation 2 is difficult to parse (what are the parameters \theta referring to? the parameters of the simple model (in which case, how does this differ from M'), or the complex model parameters?) which makes this section confusing. Additionally, is there a guarantee that Algorithm 3 converges? It would be good to include such a guarantee since (1) frames ProfWeight as an optimization problem. I wonder if your CIFAR results are influenced by the training set split? A model trained on 30,000 samples or 20,000 samples overfits quite a bit, as you note in the paper. It may be useful to try these experiments on a larger dataset like SVHN or ImageNet where a split dataset would still contain hundreds of thousands of samples. Finally, I really appreciate the real-world experiment with semi-conductor manufacturing. It is good to see a method tested in real-world settings. Smaller comments: - 16-17: some example citations would be good here - "The MAIN criticism of these models has been their lack of transparency" - this is a bold claim. - 31: "Small data settings" - It's unclear how model transfer, as you've described in this paper, would work in such a scenario. - 59: "The first function we consider, area under the curve (AUC) traced by the confidence scores, shows much improved performance." - what is the performance you're talking about? - In the description of the method (and in Algorithm 1), it may be good to use the term "layer" instead of "unit", since "unit" can mean many things in the context of neural networks. - It would be good to re-define all notation in Algorithm 1 (i.e. \beta, the arguments to \mathcal L, etc.) - Labels in 127-129 should be capitalized (e.g. as seen in Algorithm 2) - There is inconsistent notation, which makes Section 3 a bit difficult in some places. For example, the weights are a set (W) in 97, but a vector in Algorithm 1. Additionally, the inputs/labels should probably be notated as vector (e.g. in 1 - as far as I can tell - y represents all labels, and therefore should be a vector (and bolded)?) - 135: there's a reference to Line 7 of Algorithm 3, but no line numbers in Algorithm 3. Also, it would be good to repeat this step in the text, rather than making the reader refer to the algorithm block. - Algorithm 3: should define what N is - Algorithm step 4: missing a \gamma in the update weights equation? - 193: "The set of units U (as defined in Section 2.1) whose representations are used to train the probes are the units in Table 1 of the trained complex model." - I'm not sure what this means? - 196: you should explain here that set 1 is used for the complex model and set 2 is used for the simple model. I was confused by this split until I read further. - Technically, [13] did not invent distillation - [13] is a refinement of the method proposed by [9]. - You should include the following citation on your list of distillation methods: Tan, Sarah, et al. "Detecting bias in black-box models using transparent model distillation." arXiv preprint arXiv:1710.06169 (2017).

Reviewer 3



The paper aims to learn simpler classification models by leveraging a complex pre-trained model. The task is achieved by leveraging the representations learned by the intermediate layers, and learning the weights for individual data points that are then used to train a simple model. The methodology is quite intuitive and results do show the potential of the proposed scheme. However, the paper still needs to address a number of issues (detailed below). On a high level, the paper can benefit from a better positioning w.r.t. related work, a more comprehensive evaluation, and a bit more work on the clarity of the text. Detailed comments: - Line 27: The reason does not sound that compelling. Perhaps the authors want to hint at the "interpretability" of decision tree style models since one can see how the model is classifying examples (if feature_1 < a and feature_2 > b then classify as +ve). If a model (no matter how deep or "complex") provided interpretability as well as high accuracy, there might be no reason as to why domain experts would not opt for that model. - How does the technique compare to that of "Adaptive Classification for Prediction Under a Budget" from NIPS'17 (https://arxiv.org/pdf/1705.10194.pdf)? This technique also consists of training complex models, and then training simpler models to approximate the complex one in appropriate regions of the feature space. - Line 53: Are these real outputs of a network? Specifically, does the accuracy always go up as we move more towards the final layers (intuitively, it sounds like that should be the case)? - Line 69: If the simple model only focuses on the easy examples, then what happens to the hard one? Also, if would be nice to formalize how focusing on the simpler examples improves generalizability. - Line 112: What is the rationale behind training the complex model on a different dataset? Providing smaller dataset to the simpler model sounds like already setting it up for failure. - Line 118: Softmax probabilities are not necessarily well-calibrated (https://arxiv.org/abs/1706.04599). Does that make any difference on the performance of the proposed method? Also, in line 167 in the evaluation section, since the last probe is not computed in the same way as the traditional backpropagated softmax probabilities, is are the probabilities provided by these probes expected to be well-calibrated? - Table 1: The resnet paper (https://arxiv.org/pdf/1512.03385.pdf) report very different accuracy on the CIFAR-10 dataset as compared to what is observed here. Any particular reason for that? - Algorithm 2: Referring to the quantity computed here as AUC is very confusing. Are there any reasons about why this quantity corresponds to the traditional ROC-AUC? - Algorithm 3 is very difficult to understand and potentially very confusing. In step 3 of the algorithm, when computing the new weights \beta^i, why is the previous model S_{w^i-1} passed to the loss function? - Line 130: The paper notes that the weights should not all go to zero, but provides no insight as to why that might happen. Also, is there a possibility of weights becoming too high? - Section 2.2: It is not clear what part of line 7 in Algorithm 3 this section is trying to justify. Also, what precise implications do Theorem 1 and Lemma 2 have for line 7? - Line 203: Are the probes not convex? Then why not train the probes to convergence? - Line 245: Since the time since last cleaning is a feature that depends on time, wouldn't taking the last 30% samples as the test set induce some temporal bias in the evaluation? The concern might be alleviated by shedding more light on the feature "time since last cleaning" since that might affect the quality of the product. - Section 3.2: What would happen if the simple model was something other than CART. E.g., a linear SVM (which one could argue is still fairly interpretable -- specially if one were to introduce L1 regularization to zero out weights for unimportant features).